# Positive interactions within and between populations decrease the likelihood of evolutionary rescue

**Yaron Goldberg**, **Jonathan Friedman***

Department of Plant Pathology and Microbiology, The Hebrew University of Jerusalem, Rehovot, Israel

* yonatan.friedman@mail.huji.ac.il

## Abstract

Positive interactions, including intraspecies cooperation and interspecies mutualisms, play crucial roles in shaping the structure and function of many ecosystems, ranging from plant communities to the human microbiome. While the evolutionary forces that form and maintain positive interactions have been investigated extensively, the influence of positive interactions on the ability of species to adapt to new environments is still poorly understood. Here, we use numerical simulations and theoretical analyses to study how positive interactions impact the likelihood that populations survive after an environment deteriorates, such that survival in the new environment requires quick adaptation via the rise of new mutants—a scenario known as evolutionary rescue. We find that the probability of evolutionary rescue in populations engaged in positive interactions is reduced significantly. In cooperating populations, this reduction is largely due to the fact that survival may require at least a minimal number of individuals, meaning that adapted mutants must arise and spread before the population declines below this threshold. In mutualistic populations, the rescue probability is decreased further due to two additional effects—the need for both mutualistic partners to adapt to the new environment, and competition between the two species. Finally, we show that the presence of cheaters reduces the likelihood of evolutionary rescue even further, making it extremely unlikely. These results indicate that while positive interactions may be beneficial in stable environments, they can hinder adaptation to changing environments and thereby elevate the risk of population collapse. Furthermore, these results may hint at the selective pressures that drove co-dependent unicellular species to form more adaptable organisms able to differentiate into multiple phenotypes, including multicellular life.

## Author summary

Many ecosystems are exposed to rapidly changing environmental conditions, from global warming to overuse of antibiotics in medicine and agriculture. Therefore, there is great interest in elucidating the factors that affect the ability of ecosystems to adapt to these changes. While many such factors have been recently investigated, the effect of interactions within a community on its ability to adapt remains largely unexplored. In this work,

**Data Availability Statement:** All data and code files are available at https://github.com/yaronGoldberg/Evolutionary-rescue-of-positive-interactions.

**Funding:** This research was supported by the ISRAEL SCIENCE FOUNDATION (grant No. 81/097

awarded to JF). The funders had no role in study design, data collection and analysis, decision to publish, or preparation of the manuscript.

**Competing interests:** The authors have declared that no competing interests exist.

we focus on the effect of positive interactions, in the form of cooperation between individual or different species, on the ability of communities to adapt to new environments. Using simulations and theoretical analysis, we find that positive interactions significantly reduce the probability of survival of cooperative communities in changing environments, elevating the risk of populations' extinction. Furthermore, we suggest that the need for an adaptable solution of cooperation could have played a part in the circumstances leading to the transition between unicellular and multicellular life.

## Introduction

Positive interactions play key roles in shaping the assembly, function and evolution of many ecological communities [1–3]. Extensive research has demonstrated the prevalence of positive interactions in numerous ecosystems, ranging from plant communities to the human microbiome [4–10]. Positive interactions occur both as intraspecies cooperation, such as bacterial populations that are able to resist antibiotics by collectively secreting antibiotic-degrading enzymes [11,12], and as interspecies mutualism, such as the cross-protection relationship between sea anemones and clownfish [13].

Evolutionary theory demonstrates that positive interactions can be selected for. For instance, positive interactions such as nutrient exchanges between individuals, can arise due to the benefit of removing costly genes required for the generation or acquisition of the exchanged nutrients. This idea, termed 'The black queen hypothesis', is proposed as a dominant force that promotes an increase in the abundance of positive interactions between species [14,15]. Positive interactions may also be beneficial for the entire population through 'division of labour'—a situation where individuals exchange the products of different tasks in which they specialized and can perform efficiently [16]. Such benefits of positive interactions have been implicated in the evolution of multicellularity, owing to the resemblance of multicellular organisms to unicellular species that form genetically identical subpopulations of cells with different phenotypes that attain division of labour [17–19]. A well-known example of division of labour is the filamentous photosynthetic cyanobacteria that form subpopulations of nitrogen-fixing heterocysts that enable different cells to exchange the benefits of nitrogen-fixation and photosynthesis [20]. Thus, insights regarding the evolutionary dynamics of positive interactions might shed light on the transition from unicellular to multicellular life.

While the formation of positive interactions may be selected for, populations engaged in positive interactions can have heightened sensitivity to biotic and abiotic stresses. When species are interdependent, a stress that affects one species may have cascading effects that lead to the extinction of multiple additional species. This phenomenon, termed co-extinction, is found in many conservation studies examining the effects of anthropogenic environmental changes [21–23]. In addition, cooperating populations are prone to invasion by non-cooperating 'cheaters' that spread at the expense of the cooperators and can even lead to their collapse [24,25]. Such collapses occur since interactions are typically multifaceted: individuals may cooperate in one task, and simultaneously compete in another. For example, populations containing antibiotics degrading bacteria may collapse due to the rise of non-degrading cheaters that are able to outcompete the degraders for nutrients [26]. More broadly, positive interactions and competition often occur concomitantly and form a cooperation—competition continuum [27–29]. The unstable nature of positive interactions likely has significant effects on the evolution of populations and communities, but thus far research in this area has focused primarily on the sensitivity of cooperating populations to cheaters [30–32].

In particular, the effect of positive interactions on adaptation to changing abiotic environments is still poorly understood. Several recent studies suggest that adaptation may be hindered by positive interactions. First, the response of ecosystems to climate change suggests that mutualisms are unstable when adapting to novel environments [33,34]. A notable example is the cross-protection relationship between sea anemones and clownfish, that was shown to be perturbed by the imbalance of their adaptation rate [33]. Next, several experimental evolution studies have shown that abiotic stress can select against positive interactions, such as in cooperative bacterial populations grown in the presence of antibiotics [35]. In addition, studies involving bacterial mutualisms have found that while some populations evolve a more efficient division of labour and elevated growth rates, other replicate populations experience unexplained collapse, leading to extinction of both interacting species [36,37]. Finally, a recent study exploring the adaptability of metabolically co-dependent bacterial populations in the presence of antibiotics showed that co-dependency between two species results in a limitation of their adaptation ability by the least adaptable, "weakest link" strain [38,39]. Thus, once a species is more adapted to the environment than its partner, it can not further increase in fitness until it's cooperator evolves to a similar fitness. Taken together, these findings suggest that limited capacity for adaptation may be a general implication of positive interactions, but we still do not have a clear understanding of the severity of the limitations and the mechanisms causing it.

To address these questions, we focus on the extreme case of adaptation following a deterioration of a formerly hospitable environment into an inhospitable one, in which a population or community is heading towards extinction (**Fig 1A**). In this scenario, survival requires rapid adaptation to the new environment, which is possible only through the rise of new mutants—a phenomenon termed "Evolutionary rescue" [40–43]. When populations do not engage in positive interactions, it is sufficient for adapted mutants to arise prior to the ancestor's extinction

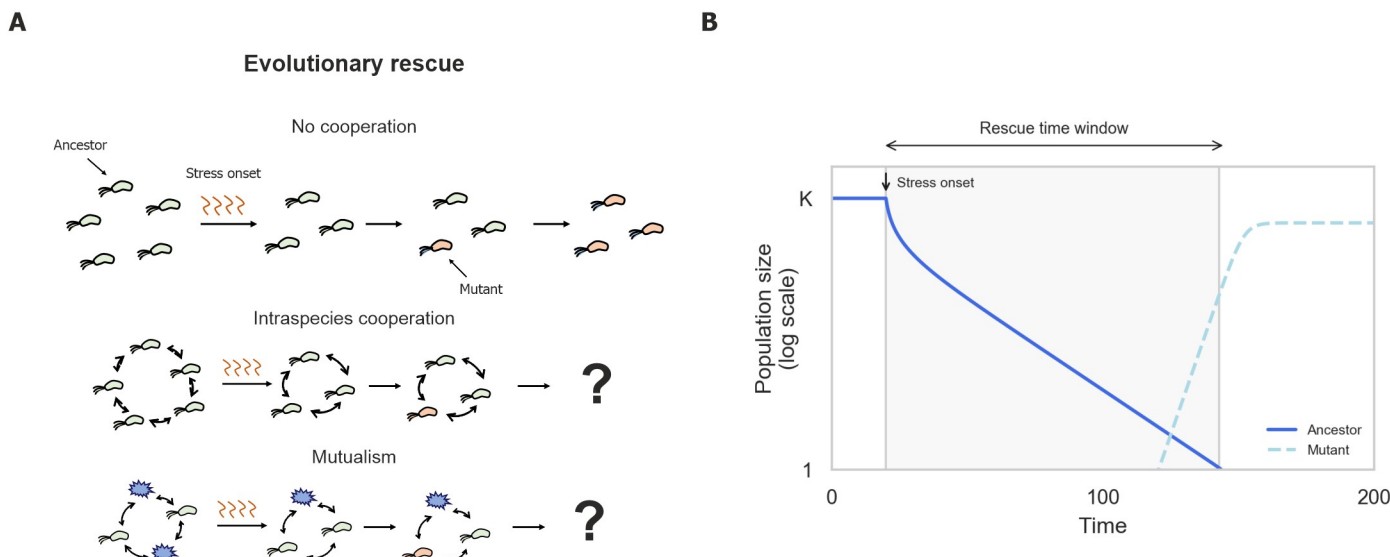

**Fig 1. Evolutionary rescue in populations engaged in positive interactions is poorly understood.** (A) An abrupt stress in the environment causes population density to decline toward extinction (green and purple cells), followed by adaptation via the rise of new mutants (orange cells). (B) Example simulation of evolutionary rescue in a non-cooperating population. Rescue time window, the time window during which adapted mutants can rise and prevent the population's extinction, is calculated as the time it takes the ancestors to become extinct. In our simulations, a population is considered extinct when it reaches below 1 individual. Parameter values used in simulations are provided in Table A in S1 Text.

in order to rescue the population (**Fig 1B**). We elucidate how different types of positive interactions influence the likelihood of evolutionary rescue by conducting numerical simulations and theoretical analyses. We conclude that while positive interactions may be beneficial in steady environments, they can hinder adaptation to changing environments. These results may offer new insights into the evolutionary dynamics of ecosystems facing sudden environmental stress, such as climate change, and the selective forces that affect cooperative and mutualistic populations over evolutionary time scales.

## Results

### Cooperative populations have a limited time window for evolutionary rescue

In order to analyze the dynamics of intraspecies cooperation when evolutionary rescue is required, we have added an evolutionary component to a previously established ecological model of cooperative populations[40] (**Fig 1,** and Eqs 1–3, and Methods, and Section B in **S1 Text**). Briefly, the model is based on the classical logistic growth model, in which populations initially grow at rate ($r$) and saturate at carrying capacity ($K$). It extends the logistic model by reducing individuals' growth rate when the population is below a critical size ($N_c$), a phenomenon known as an 'Allee effect' [44] (**S1 Fig**). For example, bacterial populations experience an Allee effect when collectively degrading antibiotics since their growth rate is increased only if there are enough degrading cells to sufficiently reduce the concentration of antibiotics in the environment [45]. In addition, we included a death rate ($\delta$) reflecting an external environmental stress that is independent of the interactions within the populations. In the example of bacteria collectively degrading antibiotics, such a stress may be a rise in temperature that impairs the bacteria's growth without affecting the antibiotic. Our model includes an ancestral population ($A$) and a mutant population ($M$) with an increased growth rate ($r_a < r_m$). The dynamics of the ancestor and mutant populations are given by:

$$\frac{dA}{dt} = r_{A(N_T)}A\left(1 - \frac{N_T}{K}\right) - \delta A \tag{1}$$

$$\frac{dM}{dt} = r_{M(N_T)}M\left(1 - \frac{N_T}{K}\right) - \delta M \tag{2}$$

The growth rate $r$ depends on the total population size (N$_T$ = A+M):

$$r_{l(N_T)} = \begin{cases} r_l & N_T > N_c \\ r_l(1-\rho) & N_T < N_c \end{cases} \quad l \in \{A, M\} \tag{3}$$

where $\rho \in [0,1]$ is the fraction by which growth rates decrease below the critical population size ($N_c$). The strong Allee effect was implemented as a step function in order to enable analytical calculations and maintain simplicity. Qualitatively similar results also occur when the Allee effect is modeled using a more complex, smooth function (**S2 and S3 Figs** and Section D in **S1 Text**).

The evolutionary rescue scenario was explored by running simulations in which an ancestor population experiences an abrupt increase in environmental death rate which causes the population to decline toward extinction, and mutants may arise stochastically during this decline, potentially spreading and rescuing the population from extinction (**Fig 1B**). Each simulation begins with the growth of an ancestral population in an unstressed environment ($\delta = 0$), followed by an onset of stress that increases the death rate such that it exceeds the ancestral

exponential growth rate ($r_A < \delta$), leading the population to decline toward extinction (**S1A Fig**). Mutation events are modeled as a Poisson process, with the expected number of mutants at each time interval given by the ancestral population size and mutation rate ($\mu$) (Eq 4 in **S1 Text**). Mutants differ from ancestors only in their elevated exponential growth rate ($r_A < r_M$), which allows them to survive the stress, but only when the total population size exceeds the critical threshold ($r_M > \delta > r_M(1-\rho)$) (**S1B Fig**). For simplicity, no further stochastic effects were considered in this model. We assessed the evolutionary rescue probability by calculating the fraction of simulations in which the mutants were able to spread and exceed the critical population size.

Consistent with previous works [46], we found that populations engaged in intraspecies cooperation have lower probability of evolutionary rescue in comparison to non-cooperative populations (**Figs 2A** and **S**8). A main cause of this reduced rescue probability is that in cooperative populations adapted mutants can spread only if they appear while the total population size exceeds the critical size, while in non-cooperative populations adapted mutants can spread regardless of the total population size. Thus, the critical population size limits the rescue time window—the time window during which adapted mutants can rise and prevent the population's extinction (**Fig 2B and 2C**). The reduction in rescue probability occurs even when cooperation provides a fitness advantage. Cooperative populations only have a rescue probability comparable to that of non-cooperative ones when their growth rate is significantly higher—up to twice that of non-cooperating populations for large critical population sizes (**S6 Fig** and Section E in **S1 Text**).

The rescue time window and rescue probability decrease as the critical population size increases. An analytical approximation of the rescue time window is given by the difference between the time it takes the ancestral population to decline to the critical population size, and the time it takes adapted mutants to grow sufficiently (Section C in **S1 Text**). The probability that an adapted mutant arises during this time interval provides an excellent approximation of the rescue probability observed in simulations (**Fig 2D**). Notably, when the critical population size is too large, mutants are unable to reach it even if they appear immediately following the onset of the stress (**Fig 2E**). Thus, in such populations the stress inevitably leads to extinction, unless adapted mutants are already prevalent enough in the population prior to the stress' onset. The mutation rate affects the likelihood of rescue when the rescue time window exists, but rescue is not possible for any mutation rate when the rescue time window is zero (**Fig 2F**). These results suggest that the higher the number of individuals required for successful cooperation, the lower the probability to adapt.

## Mutualisms have a greatly reduced probability of evolutionary rescue

Next, we test how mutualistic interactions affect the probability of evolutionary rescue. To do so, we have used an extended model in which two species are dependent on each other in an obligatory manner (Eqs 4–6, and Methods, and **S1 Text**). Analogously to the case of intraspecies cooperation, the growth rate of each species is reduced when the population size of its partner is below a critical population size ($N_c$):

$$\frac{dA_i}{dt} = r_{Ai(N_{Tj})} A_i \left(1 - \frac{N_T}{K}\right) - \delta A_i \tag{4}$$

$$\frac{dM_i}{dt} = r_{Mi(N_{Tj})} M_i \left(1 - \frac{N_T}{K}\right) - \delta M_i \tag{5}$$

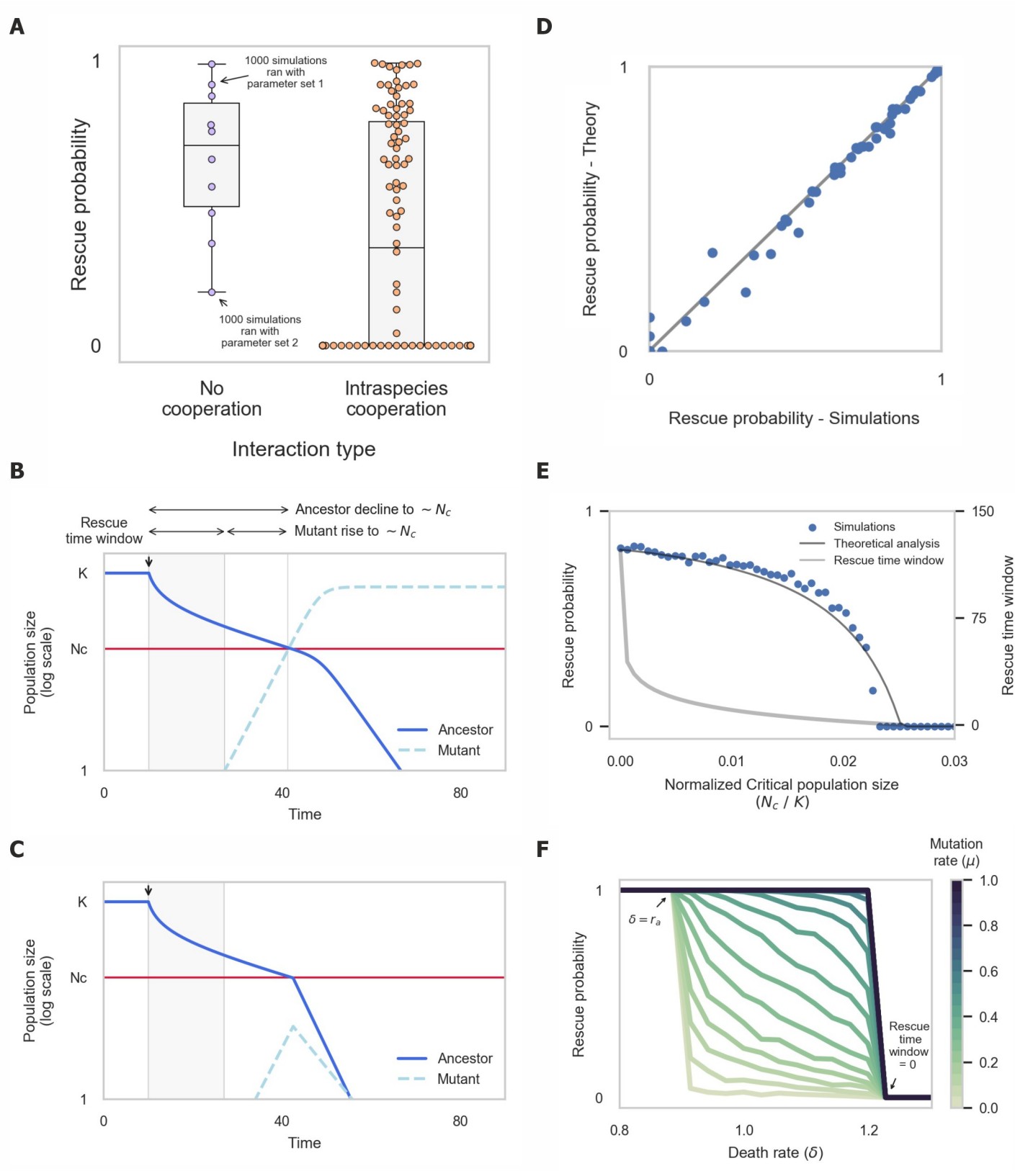

                    

**Fig 2. Cooperative populations have a limited time window for evolutionary rescue.** (A) Intraspecies cooperation has lower rescue probability in comparison to populations with no positive interactions. Each dot represents the rescue probability resulted from 1000 simulations ran with different set of parameters (Critical populations size ($N_c$), ancestor and mutant's growth rate ($r_A$,$r_M$)). (B + C) Evolutionary rescue of populations engaged in intraspecies cooperation requires the mutant's population to reach critical population size before ancestor. This results in a shorter rescue time window, outside of which mutants cannot reach critical population size and rescue the population. (D) Theoretical analysis matches well the rescue probability observed in simulations. (E) The rescue probability and rescue time window decrease as the critical population size increases. (F) Death rate ($\delta$) and mutation rate ($\mu$) effect on evolutionary rescue. Mutation rate sets the sharpness of the transition between certain rescue ($P = 1$) and certain extinction ($P = 0$).

$$r_{li}(N_{Tj}) = \begin{cases} r_{li} & N_{Tj} > N_c \\ r_{li}(1-\rho) & N_{Tj} < N_c \end{cases} \quad l \in \{A, M\}, i, j \in \{1, 2\} \tag{6}$$

Note that in this formulation, the mutualistic partners share the same resources, as captured by the fact that the total population density of both species is included in the logistic term. This represents competition for general resources, such as available nutrients, space, etc. Additionally, in our model formulation, positive interactions affect a species' exponential growth rate but not its carrying capacity (in the absence of the external death rate). This corresponds to a situation where the mutualism is important at low population densities, but at high densities growth is limited by another, independent factor. For example, when species provide each other with essential amino acids or cross-protection from antibiotics while still competing for another limited resource, such as a carbon source. Such formulation has been used in several recent works, where it agreed well with experimental results [37,47,48]. We have again assessed the rescue probability by running simulations over a range of parameters and measuring the fraction that resulted in survival of the two species. Qualitatively similar results also occur when the mutualistic interactions are modeled using a more complex, smooth function (**S4 Fig** and Section D in **S1 Text**), using different mutation rates (**S9A Fig**), when the initial densities of the species are unequal (**S9B Fig**), and when populations are also affected by their mutualistic partner at high densities (**S11 Fig** and Section F in **S1 Text**).

We observe that the probability of evolutionary rescue of populations engaged in mutualistic interactions is significantly lower than that of cooperative populations (**Fig 3A**). Since mutualistic interactions can provide fitness advantage through division of labour, we have also compared the rescue probability of populations engaged in intraspecies cooperation with that of mutualistic populations that have a higher growth rate. We found that the growth rates of mutualistic populations must be significantly greater in order for their rescue probability to be equal to that of cooperating ones, especially when critical population size is low (**S7 Fig** and Section E in **S1 Text**).

As in the case of cooperative populations, a theoretical analysis based on the length of the rescue time window approximates the probability of evolutionary rescue well (**Fig 3B** and Section C in **S1 Text**). We observed again that evolutionary rescue in mutualisms is possible only if adapted mutants arise early enough such that they are able to grow sufficiently before their partners' population declines below the critical population size (**Fig 3C**). However, this is not sufficient to explain the lower likelihood of rescue found in mutualisms compared to cooperative populations, as the duration of the rescue time window is identical in both systems.

We found two additional major effects that reduce the rescue probability in populations engaged in mutualistic interactions (**Fig 4**). First, since the two species are dependent on each other, adaptation of a single species is not sufficient to rescue it from collapse, even if the adapted mutant arises within the rescue time window (**Fig 4A and 4B**). Due to its dependence on the other species' ability to cooperate, it will collapse as soon as the second species falls below the critical population size. Thus, evolutionary rescue requires adapted mutants to arise and spread in both species in order to rescue either of them, which significantly reduces the

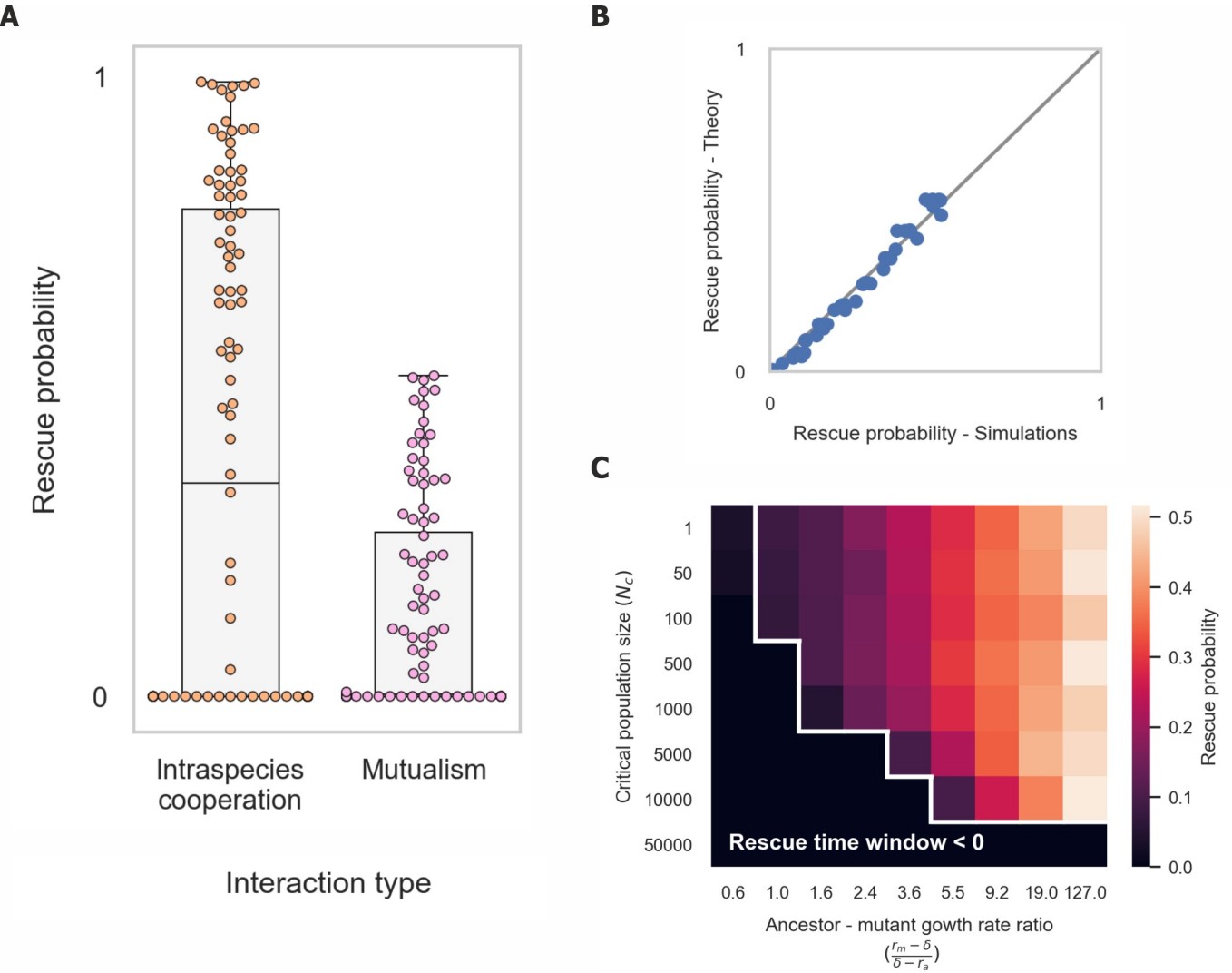

**Fig 3. Mutualisms have a greatly reduced probability of evolutionary rescue.** (A) Rescue probability is greatly reduced in mutualisms compared to intraspecies cooperation. Dots represent the rescue probability calculated from simulations ran with different sets of parameters as in Fig 2. (B) Theoretical analysis matches well the rescue probability observed in simulations. (C) Rescue probability decreases with critical population size ($N_c$) and the ratio between mutant and ancestor growth rates. As in intraspecies cooperation, the rescue time window reveals a transition curve under which rescue probability is zero.

evolutionary rescue probability since it requires the occurrence of two independent rare mutation events.

The evolutionary rescue probability is further reduced when the two species also compete (**Fig 4A and 4C**). While the two species facilitate each other's growth, they may also compete for resources, which becomes the dominant interaction at high population densities. When adapted mutants of one of the species spread and approach the carrying capacity, they outcompete their partner species for resources. This results in a faster decline of the partner species toward its critical population size and in a shortened time window for adapted mutants in this species to arise (**Fig 4C**).

Both the interdependency between species and the competition within mutualism contribute to the decline in their rescue probability. In the absence of interspecies competition mutualisms have a decreased rescue probability that is similar to that of two independent

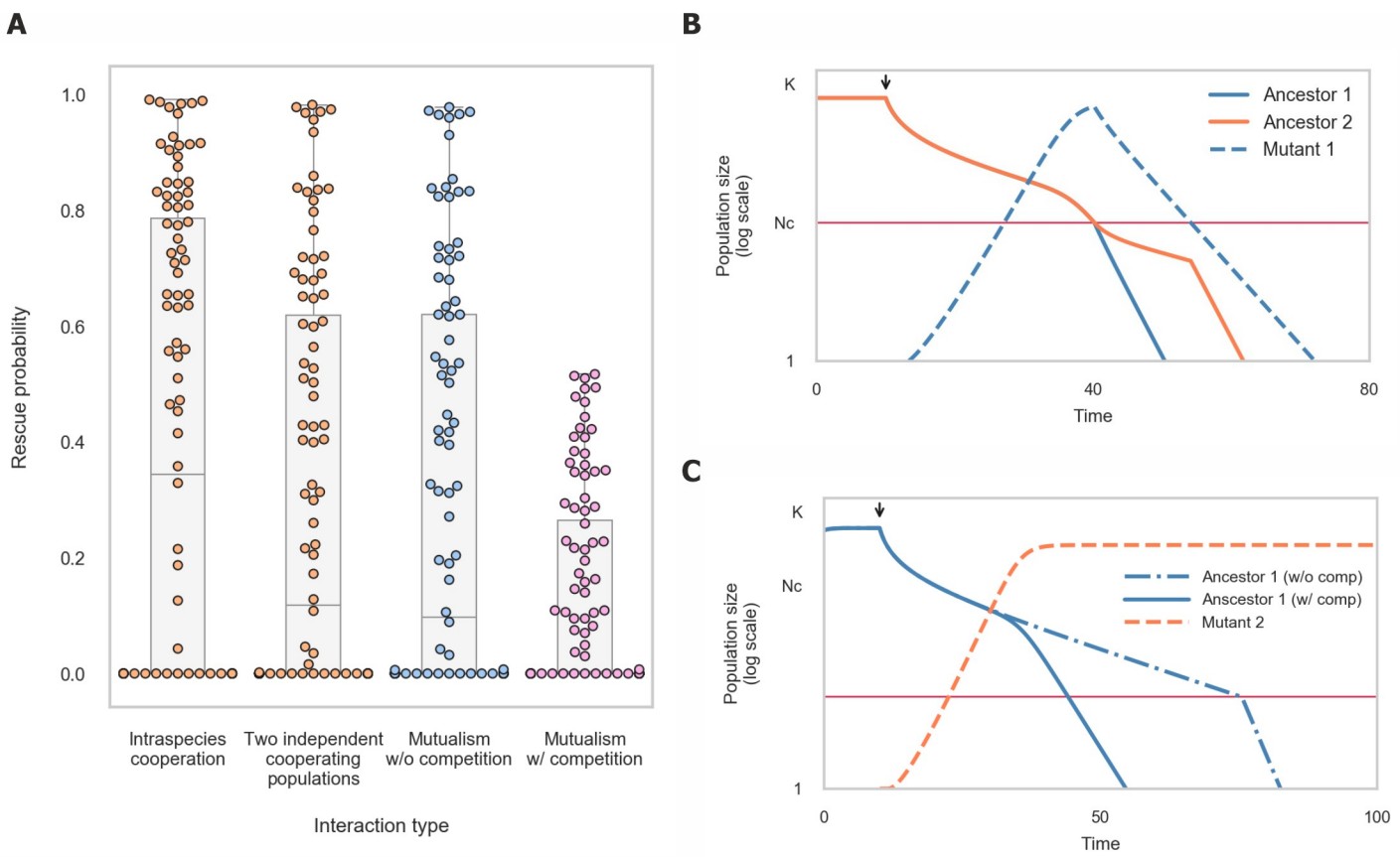

**Fig 4. Rescue probability of mutualisms is reduced since it requires adaptation of both mutualistic partners, which also compete for resources.** (A) In the absence of interspecies competition, mutualisms have a decreased rescue probability that is similar to that of two independent cooperating populations. Interspecies competition further decreases the probability of rescue of two mutualistic species. Dots represent the rescue probability calculated from simulations ran with different sets of parameters as in Fig 2. (B) Example of a simulation in which adaptation of a single species within the rescue time window does not suffice to prevent extinction due to its dependence on the other species' ability to cooperate. (C) Comparison of the rate of decline of a single species (blue) with and without competition. Adaptation of the mutualistic partner (orange) accelerates the species' decline due to competition for resources. The dynamics of the unadapted partner species are similar with and without competition and are not shown for simplicity.

cooperating populations—the rescue probability of two populations engaged in intraspecies cooperation but that do not interact with each other (since this is the rescue probability of two independent populations, it is given by the squared rescue probability of a single population). (**Fig 4A**). Interspecies competition alone decreases the probability of rescue of two non-mutualistic species, but it is the combination of competition and mutualistic interactions that jointly result in the low rescue probability found in mutualisms (**Fig 4A**). When the two mutualistic species do not compete, their rescue probability can even exceed that of two independent populations engaged in intraspecies cooperation. This occurs since adaptation of one of the species can increase the rescue time window of its partner (**S10 Fig**). However, this phenomenon only occurs for a limited set of parameters, and no parameters were found for which the influence on the rescue probability is significant.

These results suggest that mutualism may have a greatly reduced capacity for adaptation. Since different species rely on each other, adaptation of the community requires all partners to adapt. This slows down adaptation and makes it considerably less likely when limited by the supply of adapted mutations. In addition, when mutualistic partners also compete for additional resources, adaptation of one species can hinder other species' ability to adapt, potentially

leading to collapse of the whole community. We conclude that when the environment is unstable, mutualisms may be a fragile and undesirable strategy even when they offer the benefits of division of labour, as the selective pressure on species to quickly adapt to changing conditions may dominate the advantages conferred by gene loss and division of labour.

## In the presence of cheaters, evolutionary rescue of cooperative populations is extremely unlikely

Since cooperating populations are commonly invaded by cheaters, we next explored how the presence of cheaters affects the evolutionary rescue probability of cooperating populations. To do so, we adapted a previously established model that describes cooperators and cheaters dynamics and was shown to successfully capture the dynamics of yeast populations cooperating in extracellular sucrose degradation in the presence of non-degrading cheaters (Eqs 7–10) [49]. In this model, both cooperators and cheaters are affected by the cooperator's population density, and have a reduced growth rate when the cooperator population is below a critical size. However, cooperators and cheaters coexist since cheaters have a growth advantage ($b$) at high cooperator density since they do not pay the cost of cooperation, whereas at low populations densities cooperators have a growth advantage ($a$), reflecting their preferential access to the public goods they produce:

$$\frac{dA_i}{dt} = r_{Ai(N_{coop})}A_i\left(1 - \frac{N_T}{K}\right) - \delta A_i \tag{7}$$

$$\frac{dM_i}{dt} = r_{Mi(N_{coop})}M_i\left(1 - \frac{N_T}{K}\right) - \delta M_i \tag{8}$$

$$i \in \{coop, cheat\}$$

$$r_{coop,l(N_{coop})} = \begin{cases} r_l & N_{coop} > N_c \\ r_l(1-\rho) & N_{coop} < N_c \end{cases} \quad l \in \{A, M\} \tag{9}$$

$$r_{cheat,l(N_{coop})} = \begin{cases} r_l(1+b) & N_{coop} > N_c \\ r_l(1-\rho)(1-a) & N_{coop} < N_c \end{cases} \quad l \in \{A, M\} \tag{10}$$

While this model results in oscillatory dynamics, qualitatively similar results are also found in a more complex model in which oscillations do not occur (**S5 Fig** and Section D in **S1 Text**).

Invasion by cheaters reduces the cooperators' density close to the critical population size, which dramatically reduces the rescue time window and the rescue probability (**Fig 5A**). For the population to survive an adapted cooperator mutant must arise and reach sufficient population size. In contrast, adapted cheaters do not contribute to the population's growth rate and cannot prevent its extinction. Prior to the stress's introduction, the presence of cheaters causes the cooperators' population to fluctuate around the critical population size [50]. Intuitively, this happens since above the critical population size cheaters, who do not pay the cost of cooperation, have a growth advantage, whereas cooperators are at an advantage below the critical population size (e.g. due to preferential access to the public goods). This significantly shortens the rescue time window during which cooperator mutants are able reach sufficient population size before the ancestral cooperator population drops below the critical size (**Fig 5B**). Therefore, the likelihood of evolutionary rescue is greatly reduced in the presence of cheaters. In

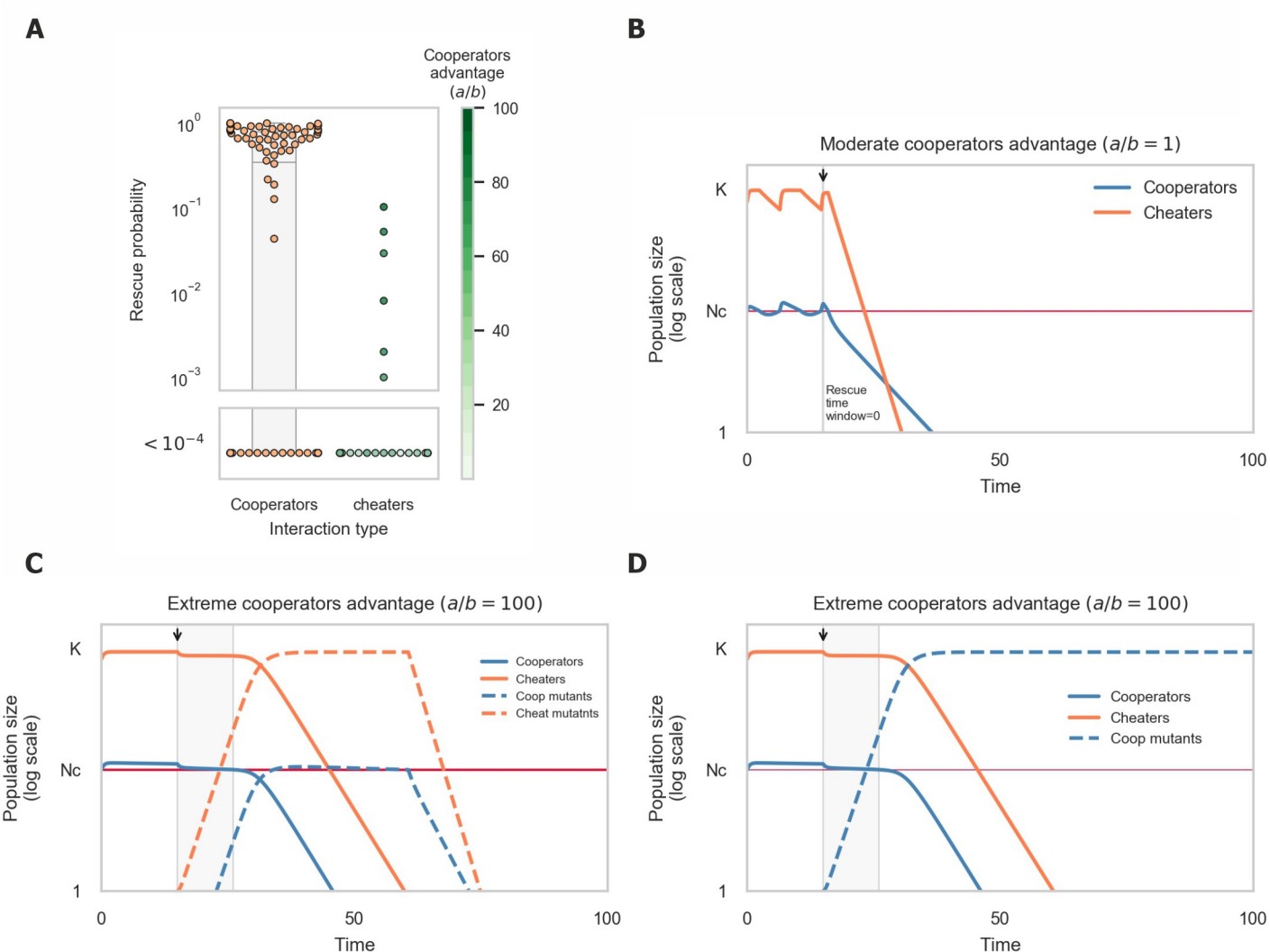

**Fig 5. Evolutionary rescue of cooperative populations in the presence of cheaters is extremely unlikely.** (A) Rescue probability of cooperative populations in the presence of cheaters is orders of magnitude lower than with no cheaters. Dots represent the rescue probability calculated from simulations ran with different sets of parameters (Critical population size ($N_c$), growth rate of ancestor and mutant ($r_A,r_M$), cooperators advantage at high densities ($a$) and cheaters advantage when cooperator is at low density ($b$)). Rescue was observed only in extreme cases, in which cooperators have a growth advantage of orders of magnitude over the cheaters when at low density (B) Cooperators oscillate around critical population size prior to the onset of stress, eliminating the rescue time window. (C+D) Cheaters are purged from populations who manage to adapt to the new environments. When cheaters manage to adapt (C), cooperators mutants are pushed below critical populations size, causing both populations to collapse. Only populations in which cooperators adapt and cheaters do not survive (D).

fact, rescue only occured in our simulations in extreme cases, in which cooperators have a growth advantage of orders of magnitude over the cheaters when at low density (**Fig 5A**).

In the extreme cases in which rescue occurs, cheaters are purged from the surviving population (**Fig 5C and 5D**). When both cheaters and cooperators manage to adapt, cooperators are rapidly pushed below critical population size due to competition with the adapted cheaters, causing both populations to collapse. This resembles the competitive effect found in mutualisms: as a population with competitive advantage adapts to the environment, interference with other populations on which it is dependent can ultimately lead to collapse of the entire system.

Our results suggest strong group selection against populations invaded by cheaters in unstable environments. Adaptation is feasible only in populations in which the cooperators'

advantage in low densities is extremely high. In addition, the fact that evolutionary rescue requires that cheaters do not adapt to the new environment constitutes a selective pressure towards purging of cheaters from the population.

## Discussion

Our findings reveal that positive interactions can significantly decrease populations' likelihood of evolutionary rescue (**Fig 6A**). We found that this reduction is mainly due to the fact that survival in such populations requires at least a minimal number of cooperating individuals, reducing the time window during which adapted mutants can rise and spread. In mutualistic populations, we observed that the reduction of rescue probability is exacerbated by two additional effects: First, due to codependency between the two mutualistic partners, the rise and spread of adapted mutants in each of the populations is required in order to prevent either of the populations from collapsing. Second, due to competition for resources, adaptation of one of the species accelerates the decline of its partner towards extinction. Finally, we demonstrated that the presence of cheaters reduces the likelihood of evolutionary rescue even further, making it extremely unlikely, primarily since the cooperator population is close to its critical population size prior to the onset of the stress.

Since populations engaged in positive interactions are more prone to collapse due to their reduced capacity for evolutionary rescue, how can they be so prevalent in nature? One possibility is that positive interactions occur primarily in steady environments, where frequent adaptation via evolutionary rescue is not essential. In addition, positive interactions may occur transiently—arising when the environment is stable due to selection for gene loss or division of labour, and collapsing when conditions change dramatically and adaptation is required. Positive interactions may also be more stable when they occur as mutualism between species that do not compete strongly, since such competition greatly reduces mutualisms' evolutionary rescue probability (**Fig 6A**). For example, little or no competition occurs between plants and their mutualistic pollinator partners [51,52]. Lastly, spatial structure, where individuals can interact preferentially with their neighbors and kin has been shown to promote positive interactions and may increase the likelihood of evolutionary rescue [53–55].

In changing environments, where severe stresses occur periodically, there can be a conflict between maintaining the capacity for adaptation and gaining the benefits of division of labor (**Fig 6B**). Mutualism offers the benefits of division of labour but has low adaptability. In contrast, generalism, where each individual performs all the tasks required for its growth, has high adaptability but does not provide the benefits of division of labour. A strategy that can offer the benefits of division of labour while maintaining the capacity for adaptation is phenotypic variability—a single genotype that differentiates into several specialized phenotypes [19,56,57]. In this situation, division of labour is enabled by positive interactions between different specialized phenotypes, while the capacity for adaptation is maintained since these phenotypes share a single genome. Thus, both the interacting phenotypes can adapt via a single adaptive mutation, and the competitive effect that occurs during the adaptation of two mutualistic species is circumvented.

Our observation that the presence of cheaters significantly lowers populations' ability to adapt underscores the importance of cheater avoidance mechanisms when facing environmental changes. Prevention of cheaters invasion is commonly achieved through population spatial structure, which enables cooperators to preferentially interact with each other [7,58–60]. Therefore, in changing environments, the combined selective pressure imposed by the need to divide labour, adapt to novel environments, and limit exploitation by cheaters may select for the formation of populations that differentiate into individuals with specialized phenotypes

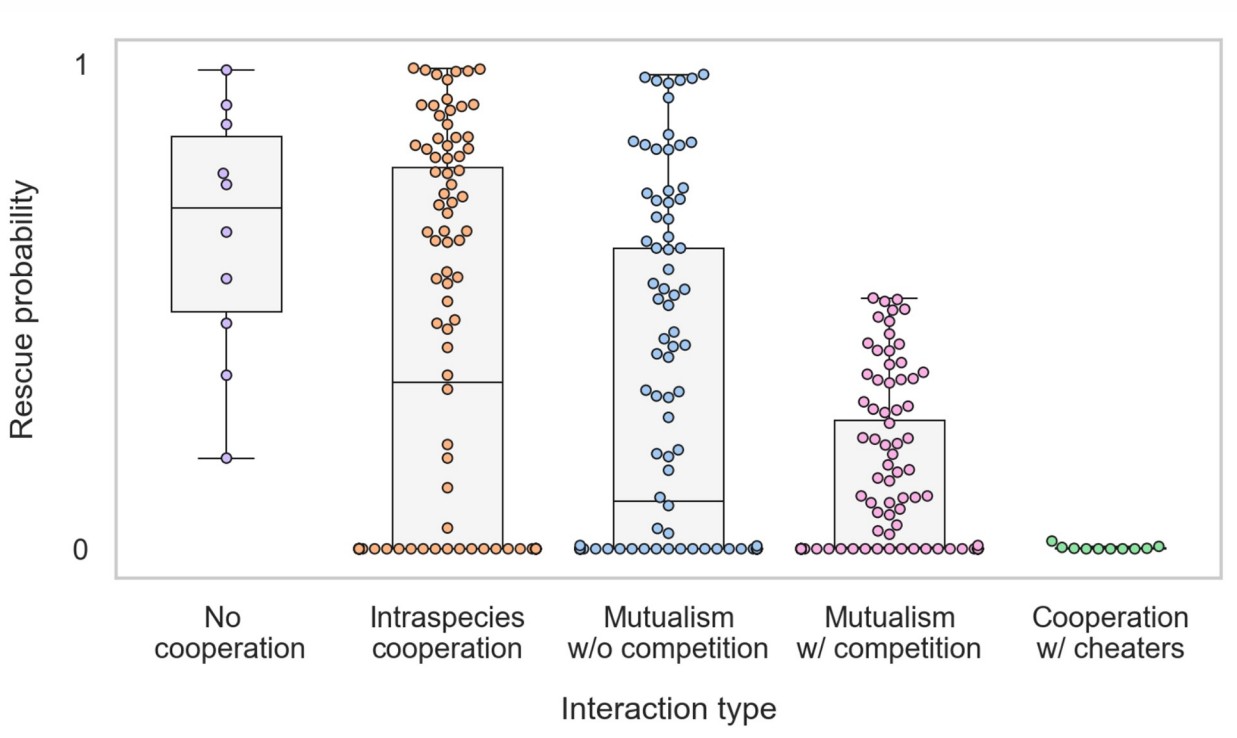

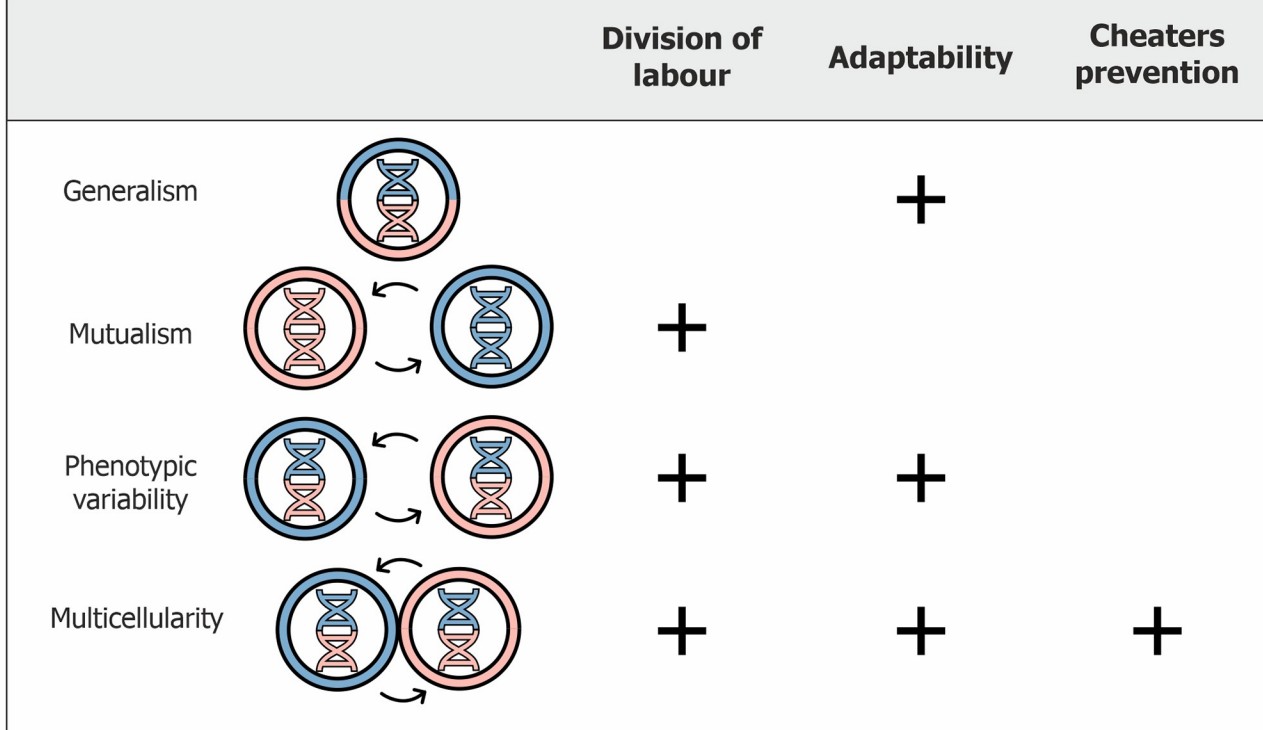

**Fig 6. The reduced adaptability of positive interactions may have contributed to the formation of more complex strategies for division of labour.** (A) Summary of observed rescue probability for different types of interactions. Dots represent the rescue probability that resulted from simulations ran with

different sets of parameters as in Fig 2. (B) Different strategies of cooperation and the selective advantage they permit. The combined selection pressure imposed by the need to divide labour, adapt to novel environments and limit exploitation by cheaters may select for the formation of populations that differentiate into individuals with specialized phenotypes that form tight spatial structures, and potentially even multicellular organisms. Colors represent two different tasks required for growth, helix and circle colors represent genotype and phenotype, respectively.

that form tight spatial structures (**Fig 6B**). Strikingly, several organisms that are thought to be precursors to true multicellularity, such as choanoflagellates and volvox, fulfil these criteria [47–50]. More broadly, the elevated risk of extinction experienced by cooperative groups invaded by cheaters generates strong selection against cheaters at the group level [61–63]. Such group or lineage selection is considered a key factor leading to the evolution of multicellular organisms [64,65]. Thus, our insights into the influence of positive interactions on adaptation suggest new selective forces that might have been involved in the transition between unicellular and multicellular life.

Our results demonstrate the importance of considering ecological interactions when addressing evolutionary questions. Research interest in species adaptation to new environments has rapidly increased in recent years due to the realization that human activities are causing major changes to the environment of numerous ecosystems. Our findings highlight the potential of interactions within ecosystems to alter their fate in the face of environmental changes, such as those caused by anthropogenic influences.

## Methods

### Numerical simulations

We have constructed simple models to describe the dynamics of positive interactions when evolutionary rescue is required, which is detailed in the supporting information (**S1 Text**). Briefly, the models extend the logistic growth rate by applying strong Allee effect and addition of external growth rate (Eqs 1–10 and **S1 Text**). Each simulation began with the growth of an ancestral population in an unstressed environment ($\delta = 0$), followed by an onset of stress that increases the death rate such that it exceeds the ancestral exponential growth rate ($r_A < \delta$), leading the population to decline toward extinction. Simulations were ran in discrete time intervals ($\Delta t = 0.01$), and the number of mutants $M$ that arose during a time interval was sampled as a Poisson process, with the expected number of mutants determined by the ancestral population size and the mutation rate (Eq 4 in **S1 Text**). Throughout the simulations, populations whose size decreased below 1 were considered to be extinct. In addition, to maintain simplicity, no further stochastic effects were considered in this model. Simulations end when extinction or rescue occurs, as defined for each model in the supporting information. Evolutionary rescue probability was calculated by running 1000 simulations for each parameter set (Table A in **S1 Text**), and calculating the fraction of simulations that resulted in rescue. The simulations were implemented using custom python scripts, using Scipy integrator for the calculation of population size change within each time step. The code is available upon request.

## Supporting information

**S1 Text. Supporting information including modeling, theory and further information.** Section A: Modeling Assumptions. Section B: Models. Section C: Theoretical analysis. Section D: Models with continuous Allee effect. Section E: Positive interactions provide a fitness advantage. Section F: Model of mutualism in which populations are affected at high densities. Table A: Parameters table.
(DOCX)

**S1 Fig. Change in per capita growth rate of intraspecies cooperating populations.** (A) Ancestor per capita growth rate as a function of total population size after stress onset. Growth rate decreases with population size due to intraspecies competition. When the population size is below the critical population size ($N_C$), the growth rate reduces further due to the Allee effect. Ancestor growth rate is always negative due to environmental stress. (B) Mutant per capita growth rate as a function of total population size after stress onset. Here, growth rate can be positive above ($N_C$) due to higher growth rate ($r_M$) Thus, survival is possible when the total population size is sufficiently high.
(TIF)

**S2 Fig. Change in per capita growth rate of intraspecies cooperating population implemented with a continuous model.** (A) Ancestor per capita growth rate as a function of total population size after stress onset. When population size is above critical population size ($N_C$), growth rate decreases as population size increases due to carrying capacity (K). When below ($N_C$), growth rate reduces further due to Allee effect. Ancestor growth rate is always negative due to environmental stress. (B) Mutant per capita growth rate as a function of total population size after stress onset. Here, growth rate can be positive when above ($N_C$), thus adaptation is possible when total population size is sufficiently high.
(TIF)

**S3 Fig. The continuous model of cooperative populations shows qualitatively similar behaviour to that of the discontinuous model.** (A) Intraspecies cooperation has lower rescue probability in comparison to populations with no positive interactions, similarly to the discontinuous model. The median of the continuous model is lower since rescue probability decreases when critical populations size increases. Each dot represents the rescue probability resulted from 1000 simulations ran with different set of parameters (Critical populations size ($N_C$), ancestor and mutant's growth rate ($r_A,r_M$)). (B) The rescue probability decreases as the critical population size increases. (C) Rescue probability decreases with critical population size ($N_C$), and the ratio between mutant and ancestor growth rates.
(TIF)

**S4 Fig. The continuous model of mutualistic populations shows qualitatively similar behaviour to that of the discontinuous model.** (A) Rescue probability is greatly reduced in mutualisms compared to intraspecies cooperation, similarly to the discontinuous model. Dots represent the rescue probability calculated from simulations ran with different sets of parameters as in S3 Fig. (B) Rescue probability decreases with critical population size ($N_C$) and the ratio between mutant and ancestor growth rates. (C) Death rate ($\delta$) and mutation rate ($\mu$) effect on evolutionary rescue.
(TIF)

**S5 Fig. Evolutionary rescue of cooperative populations in the presence of cheaters implemented by a continuous model is extremely unlikely.** (A) Rescue probability of cooperative populations in the presence of cheaters is orders of magnitude lower than with no cheaters, similarly to the discontinuous model. Rescue in the continuous model was observed only in extreme cases, even more than the discontinuous model, in which cooperators have a growth advantage of orders of magnitude over the cheaters when at low density. (B+C) The dynamics observed between the cooperators and cheaters are not oscillatory. When $b>>a$ (B), rescue is not possible since cooperators density is below critical population size. Evolutionary rescue is only possible if $a>>b$ (A) when cooperators density prior to stress is above critical population size.
(TIF)

**S6 Fig. Cooperative populations have a rescue probability comparable to that of non-cooperative ones only when their growth rate is significantly higher.** (A) Ratio of the growth rate of cooperating and non-cooperating populations in which their evolutionary rescue probability is equal. At low critical populations size, the ratio is 1 since cooperation does not affect the evolutionary rescue probability. As critical population size increases, the ratio increases since the rescue time window decreases. At a high growth rate ratio, the evolutionary rescue matches only when both populations have no chance of rescue. (B) The same analysis for a wider parameters range. The ratio for which the evolutionary rescue probability of the two populations matches increases with the ratio between the growth rate of the mutant and ancestor, up to a point in which the growth rate of non cooperating species is twice that of non-cooperating populations for large critical population sizes.
(TIF)

**S7 Fig. Evolutionary rescue probability of mutualism and intraspecies cooperation equals when mutualism growth rate is significantly higher.** (A) Ratio of the growth rate of populations engaged in mutualism and interspecies cooperation in which their evolutionary rescue probability matches. As opposed to comparison with non-cooperating populations, the ratio decreases with critical population size. At low critical population size, mutualisms must have a high growth rate advantage due to the requirement for two mutations and due to competition. At high critical population size, both rescue probabilities decrease to zero at the same critical population size due to equal limited rescue time window. (B) The same analysis for wider parameters range. The ratio for which the evolutionary rescue probability matches increases with the ratio between the growth rate of the mutant and ancestor.
(TIF)

**S8 Fig. Cooperative populations have a limited time window for evolutionary rescue.** (A) Comparison of the rescue time window (grey) of non cooperating populations (solid line, light grey) and populations engaged with interspecies cooperation (dashed line, dark grey). (B) Rescue probability decreases with critical population size ($N_C$) and the ratio between mutant and ancestor growth rates. The theoretical rescue time window (white line) reveals a transition curve under which the rescue probability is zero.
(TIF)

**S9 Fig. Mutualism evolutionary rescue probability as a function of different parameters.** (A) Death rate ($\delta$) and mutation rate ($\mu$) effect on evolutionary rescue probability of mutualistic populations. (B) Evolutionary rescue probability as a function of the fraction of initial population size of one species. Rescue probability decreases as one of the mutualistic partners begins at lower initial density.
(TIF)

**S10 Fig.  In limited conditions, adaptation of one of the species can increase the rescue time window of its partner.** An example of simulation in which adaptation of one of the species (blue) increases the rescue time window of its partner (orange). Since the adapted mutant spreads and reaches critical population size, the growth rate of its cooperator is not impaired. Thus, the second mutation event can happen at any time point prior to the ancestors' extinction.
(TIF)

**S11 Fig. Rescue probability of populations engaged in mutualism, where positive interactions affect populations at high densities, shows qualitatively similar results to that of the**

**original model.** Dots represent the rescue probability calculated from simulations ran with different sets of parameters as in Fig 2 in the main text.
(TIF)

## Acknowledgments

We thank Nadav Kashtan and Alfonso Pérez Escudero for constructive comments on the manuscript, as well as Aswin Krishna and the members of the Friedman lab for helpful discussions.

## Author Contributions

**Formal analysis:** Yaron Goldberg.

**Funding acquisition:** Jonathan Friedman.

**Investigation:** Yaron Goldberg, Jonathan Friedman.

**Supervision:** Jonathan Friedman.

**Visualization:** Yaron Goldberg.

**Writing – original draft:** Yaron Goldberg.

**Writing – review & editing:** Jonathan Friedman.

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
