## [Decision Letter · Decision Letter 0]

4 Nov 2020

Dear Dr Friedman,

Thank you very much for submitting your manuscript "Positive interactions within and between populations decrease the likelihood of evolutionary rescue" for consideration at PLOS Computational Biology. As with all papers reviewed by the journal, your manuscript was reviewed by members of the editorial board and by several independent reviewers. The reviewers appreciated the attention to an important topic. Based on the reviews, we are likely to accept this manuscript for publication, providing that you modify the manuscript according to the review recommendations.

Sincerely,

Jacopo Grilli

Associate Editor

PLOS Computational Biology

Stefano Allesina

Deputy Editor

PLOS Computational Biology

[LINK]

Reviewer's Responses to Questions

**Comments to the Authors:**

Reviewer #1: The authors set up an in silico investigation of the influence of positive interactions on a populations chance of rescue in a harsh environment. The model is based on a logistic growth, modified to incorporate the Allee effect and environment-dependent death rate. The rescue assay they have set up is also intuitive and suitable for the focal question. Overall, the paper is well-written, has a logical flow, and is accessible in my opinion to a broad audience.

In my view, the paper adds to the current understanding of the communities at the intersection of ecology and evolution and would be suitable for publication after addressing the following comments:

1. The main findings of the paper appear to be resulting from enforcing the Allee effect on small populations, which disproportionately impacts positive interactions. It is possible that positive interactions do not follow this model, especially in a structured environment (see for example (Chao & Levin, PNAS, 1981) for intraspecies cooperation or (Momeni et al, eLife, 2013) for interspecies cooperation)

2. I have concerns about Eq. S6 and S7. In mutualism, conventionally the growth rate of each species is set to be modulated by the population size of the other species. However, in the current formulation, the rate of change in populations is proportional to the population size of the partner. The current formulation would perhaps make sense if the mutualism is based on exchange of consumable mediators. I think the justification for this formulation should be included in the manuscript. Also, if not overly burdensome, I recommend confirming the results using a Lotka-Volterra type formulation as well.

3. In mutualism results, the assumption is that the environmental stress affects both populations. This makes it somewhat trivial that the rescue is less likely for mutualism compared to a monoculture rescue. It would be interesting to test what happens if the stress affects only one of the populations. In other words, is the rescue less or more likely for a species in an abiotic environment versus in a biotic environment of its mutualistic partner.

4. In studying the impact of cheaters, the authors are focusing on a situation that the stable population ratio of the community is heavily in favor of the cheater. Do you expect the same results if the stable community composition contained a cooperator majority and a minority of cheaters?

Reviewer #2: To understand the effects of positive interactions on the ability to adapt to a stress, Goldberg and Friedman performed numeric simulations mimicking microbial growth in a flask (no spatial structure). The authors modeled two types of positive interactions: intraspecies cooperation and interspecies mutualism. First, they modeled cooperation by modifying the logistic growth model to include the Allee effect which caused the growth rate to slow below a critical population threshold (Nc). To avoid extinction it was critical that the population remained above the Nc. At the beginning of the simulations, an ancestral population was exposed to a stress that causes them to rapidly die (the death rate exceeded the growth rate). During this period, mutants were modelled to arise as a Poisson process. These mutants had an increased growth rate and could therefore survive the stress if they reached a population size above the Nc. The mutation rate and ancestral population size affected the likelihood of mutants arising at any given time. These mutants were dependent on the population remaining above the Nc or else they too would experience the Allee affect and go extinct. Not surprisingly, the authors found that non-cooperators -- populations that did not experience the Alee effect -- had a higher probability of survival than the cooperator populations. When the populations (both cooperative and non-cooperative) survived the stress it was due to mutants arising within the ‘rescue time window’. Secondly, the authors modeled mutualism in which a species’ growth rate was dependent on the other species’ population remaining above the critical population size in order to avoid the Allee effect. The authors found that under mutualistic conditions, species had a higher probability of extinction than under a cooperative scenario because mutants in both species needed to arise within the rescue time window. Unsurprisingly, cheaters and competition further reduced the probability of stress survival.

Overall, this paper advances the discipline and sets the stage for future experimental work.

Major Comments: NA

Minor Comments:

Throughout the manuscript there are many typographical errors. In figure 2A, “parameter” is misspelled. In figure 3C, “Ancestor” is misspelled. Furthermore, there are many additional typos in the supplementary material and code repository. Please proofread and correct these mistakes..

Figure 2E was initially confusing to me. Please include a label for the ‘Rescue time window’ in the symbol key in the top right corner.

Lines 122-124: Provide a citation for this statement.

Lines 248-252: Provide the equation #s that describes competition for resources in the mutualism.

Lines 257-259: Provide a more detailed description of the ‘two independent cooperating populations’. This was initially unclear to me.

There may be an issue in the jupyter notebook, Models.ipynb, as it does not work for me. I’m not sure if this is an issue on my end or if there is an error in the code. This is the error:

NameError Traceback (most recent call last)

<ipython-input-20-abcf49327300> in <module>

-- 1 def simulate_cheaters(model=derivative,t=t,dt=dt,mu=mu,Nc=Nc,N1_initial=N1_initial,N2_initial=N2_initial,plot_bool=False,r1=r1,r2=r2,rm1=rm1,rm2=rm2,ro1=ro1,ro2=ro2,d_initial=d_initial,d_stress=d_stress,a=a,b=b,stress_onset=stress_onset,ax=None):

2 '''

3 Runs simulation of mutualism

4 model - The derivative

5 t - time of simulation

NameError: name 'ro1' is not defined</module></ipython-input-20-abcf49327300>

**Have all data underlying the figures and results presented in the manuscript been provided?**

Reviewer #1: Yes

Reviewer #2: Yes

PLOS authors have the option to publish the peer review history of their article (what does this mean?). If published, this will include your full peer review and any attached files.

Reviewer #1: No

Reviewer #2: No
---

## [Decision Letter · Decision Letter 1]

21 Jan 2021

Dear Dr Friedman,

We are pleased to inform you that your manuscript 'Positive interactions within and between populations decrease the likelihood of evolutionary rescue' has been provisionally accepted for publication in PLOS Computational Biology.

Best regards,

Jacopo Grilli

Associate Editor

PLOS Computational Biology

Stefano Allesina

Deputy Editor

PLOS Computational Biology

Reviewer's Responses to Questions

**Comments to the Authors:**

Reviewer #1: The authors have addressed all the comments. I have no further concerns and recommend the manuscript for publication.

Reviewer #2: The revisions for this paper satisfactorily addressed my previous concerns. Very nice job.

**Have all data underlying the figures and results presented in the manuscript been provided?**

Reviewer #1: Yes

Reviewer #2: Yes

PLOS authors have the option to publish the peer review history of their article (what does this mean?). If published, this will include your full peer review and any attached files.

Reviewer #1: **Yes: **Babak Momeni

Reviewer #2: No

---

## [Editor Report · Acceptance letter]

11 Feb 2021

PCOMPBIOL-D-20-01712R1 

Positive interactions within and between populations decrease the likelihood of evolutionary rescue

Dear Dr Friedman,

I am pleased to inform you that your manuscript has been formally accepted for publication in PLOS Computational Biology. Your manuscript is now with our production department and you will be notified of the publication date in due course.

With kind regards,

Alice Ellingham
